# Application of Superhydrophobic Mesh Coated by PDMS/TiO_2_ Nanocomposites for Oil/Water Separation

**DOI:** 10.3390/polym14245431

**Published:** 2022-12-12

**Authors:** Kun Cao, Xi Huang, Jie Pan

**Affiliations:** College of Chemistry and Chemical Engineering, Neijiang Normal University, Neijiang 641100, China

**Keywords:** superhydrophobic, coating, TiO_2_ nanofiber, oil–water separation

## Abstract

Superhydrophobic materials have recently attracted great interest from both academia and industry due to their promising applications in self-cleaning, oil–water separation, etc. Here, we developed a facile method to prepare hybrid PDMS/TiO_2_ fiber for superhydrophobic coatings. TiO_2_ could be uniformly distributed into PDMS, forming a hierarchical micro/nano structure on the surface of the substrate. The contact angle of the superhydrophobic coating could reach as high as 155°. The superhydrophobic coating possessed good self-cleaning performance, corrosion resistance, and durability. It was found that gravity-driven oil–water separation was achieved using stainless steel mesh coated with the PDMS/TiO_2_ coating. More importantly, the coated filter paper could not only separate oil and pure water but also corrosive solutions, including the salt, acid, and alkali solution.

## 1. Introduction

With the development of the economy and industries and the improvement of people’s living standards, environmental issues are increasingly of concern. Especially the oily wastewater produced by industrial emissions, accident leakage, domestic waste, etc., not only pollutes water but also threatens human life. Furthermore, the discharge of oils also leads to a great loss of energy and resources. Thereby removal of oil from wastewater is challenging work for the conservation of ecosystems. Various cleanup methods based on bioremediation, chemical treatment, and mechanical recovery have been applied to the separation of oil–water mixtures [1,2,3,4,5,6,7,8]. The design and preparation of superhydrophobic surfaces with water contact angles >150° have garnered widespread attention in terms of basic research and industrial applications [9,10,11,12]. Nature gives good examples, such as the lotus leaf with self-cleaning properties due to a bumpy waxy surface with hierarchical micro/nanostructures. High surface roughness and low surface free energy are necessary prerequisites for obtaining stable superhydrophobic characteristics (water contact angle of more than 150° and sliding angle of less than 10°). Metal mesh shows great potential due to its high thermal and mechanical stability. In some studies, nanoparticles are endowed with superhydrophobic properties on the mesh surface by etching and deposition [13,14], but such methods will significantly reduce the mechanical properties of the substrate under certain harsh environments, thus limiting their application. In other studies, nanoparticles are combined with polymers, and polymer resins are used as binders to prepare superhydrophobic and superhydrophilic coatings so as to bind nanoparticles to the mesh surface [15,16,17,18]. Cao [16] spray-modified a PU/SiO_2_ composite coating to make the grid superhydrophobic to hot water. In another study, PMMA was used to improve the strength between carbon nanotubes and steel mesh surfaces [17].

In recent years, the fabrication of superhydrophobic surfaces based on semi-conductor oxide materials such as ZnO [18], CuO [19], and TiO_2_ [20] has been conducted. TiO_2_ nanostructures have become a focus of tremendous interest due to their unique physicochemical properties and good chemical stability related to applications including photocatalysis [21], anti-ultraviolet protection [22], and chemical sensors [23]. Nanosized TiO_2_ powder is now commercially available and can be used as a raw material for the preparation of superhydrophobic surfaces. Bolvardi et al. [24] reported commercialized TiO_2_ nanoparticles with different sizes forming a coating consistent with PDMS. Qing et al. [25] fabricated superhydrophobic surfaces of TiO_2_/PDMS by dip coating, and the as-prepared superhydrophobic surface had an excellent self-cleaning ability. 

Compared with the spherical structure, the top size of the cluster-like structure was larger, and the size of the bottom was smaller. Capillary pressure, namely, the required pressure yielded by the air–water interfacial area to resist water into the micro–nano structure gaps, decreased with increases in the nanocrystal spacing. The cluster-like TiO_2_ nanofiber spacing was smaller than the TiO_2_ spherical structure. Therefore, when undergoing oil–water separation recycling, TiO_2_ nanofibers had better separation performance [26]. In this paper, TiO_2_ nanofibers prepared by the hydrothermal method were combined with PDMS to prepare a composite coating, which was sprayed on the surface of stainless steel mesh to form a fiber cluster. Moreover, the effects of mechanical friction, the pH of the mixture, and continuous oil–water separation performance of superhydrophobic stainless steel mesh were studied. This method can rapidly and effectively prepare oil–water separation materials on a large scale and can be used in harsh environments.

## 2. Materials and Methods

### 2.1. Preparation of TiO_2_ Nanofibers

One gram of TiO_2_ (P25) was added to 30 mL, 11 mol/L NaOH, stirred evenly, and dispersed ultrasonically for 30 min and then added to a 50 mL steel-lined autoclave for hydrothermal treatment at 170 °C for 24 h. After cooling to room temperature and washing with distilled water for several times for neutralization, a white solid was obtained by centrifugation and vacuum dried at 50 °C for 12 h to obtain sodium titanate nanofibers. The sodium titanate nanofibers were soaked in 0.1 mol/L hydrochloric acid solution for 24 h, washed again with distilled water to neutralize, centrifuged to remove the supernatant, dried in vacuum at 50 °C for 12 h, and sintered at 500 °C for 2 h to obtain TiO_2_ nanofibers.

### 2.2. Preparation of PDMS/TiO_2_ Composite Coating

The preparation of superhydrophobic stainless steel mesh is shown in Figure 1. The PDMS main agent and curing agent were mixed in a ratio of 10:1, 1 g of the mixture was dissolved in 10 g of n-hexane, and the mixture was ultrasonically dispersed for 10 min. Then, TiO_2_ nanofibers with a mass fraction of 1% were added to the mixture and ultrasonically dispersed for 10 min to prepare the PDMS/TiO_2_ composite coating. The dispersed composite coating was directly sprayed onto glass slides, stainless steel sheets (SSS), and stainless steel mesh (SSM) and cured at 120 °C for 40 min, which were used for tests of self-cleaning performance, corrosion resistance, and oil–water separation performance, respectively.

### 2.3. Sample Testing and Characterization

The crystal nature and surface morphology of TiO_2_ nanofibers prepared by the hydrothermal method were characterized by an Ultima IV X-ray diffractometer and a JEM-2010 high-resolution transmission electron microscope. The surface morphology of the coated sample was studied by scanning electron microscopy (SEM) (TEDCAN VEGA 3 SBH). The wettability of the SSM was measured via contact-angle using an SDC-350 optical contact angle measuring instrument with a 5 μL distilled water droplet at room temperature. The contact angle values were averages of three independent measurements at ambient temperature.

### 2.4. Anti-Corrosion Property Test

The corrosion resistance of the superhydrophobic coating was tested by electrochemical impedance spectroscopy (EIS). The corrosion medium was 3.5% NaCl solution. A three electrode test system was adopted, the coated SSS was used as the working electrode, and the platinum sheet and saturated calomel electrode were used as the counter electrode and the reference electrode respectively. The scanning frequency range was 0.01 Hz to 10^5^ Hz, and the amplitude was 10 mV.

### 2.5. Oil–Water Separation Performance Test

Distilled water was used as the water phase, methylene blue was used for dyeing, and hydrochloric acid, sodium hydroxide, and sodium chloride were added, respectively, to test the chemical resistance properties. N-hexane, chloroform, vegetable oil, and mineral oil were selected as the oil phase (and oil red O was used for dyeing) and mixed with the water phase in a 1:1 ratio to form an oil–water mixture. The coated SSM were fixed on the self-made oil–water separation device. The oil–water mixture flowed slowly to the coated SSM and was separated by gravity. 

The separation efficiency was calculated by Equation (1):*η* = *V*_2_/*V*_1_ × 100%(1)
where *η* is the oil–water separation efficiency, and *V*_1_ and *V*_2_ are the volume of oil phase before and after separation, respectively.

### 2.6. Durability Test

The durability test included two aspects, one to test the wear resistance of the coated SSM, and the other to test the impact of the number of cycles on the separation efficiency. The wear resistance of the coated SSM was characterized by the contact angle after friction on sandpaper. The test surface of coated SSM was placed on #200 sandpaper, and a 100 g weight was added. The test piece was horizontally pushed to travel 15 cm and then vertically pushed to travel 15 cm as a cycle; the contact angle was tested after every 5 cycles. A water/n-hexane mixture was used as the test system for the circulation of the coated SSM, and the separation efficiency was measured for 50 cycles.

### 2.7. Self-Cleaning Test

In order to verify the self-cleaning performance of the coated SSM, the soil was used as the pollutant. The slide with the composite coating was placed against a small-angle-inclined circular dish. A thin layer of soil powder was distributed on the surface, and then a syringe was used to clean the surface. During this process, photos were taken at different times.

### 2.8. Water Jet Stability

The water jet stability was tested by spraying water toward the coated SSM, and the entire process was recorded with a camera. The microstructure and the contact angle after the water jet test was also characterized.

## 3. Results and Discussion

### 3.1. Morphology and Composition of TiO_2_ Nanofibers

Figure 2 shows the TEM morphology and XRD diagram of TiO_2_. It can be seen from Figure 2a,b that the TiO_2_ powder was prepared into smooth and uniform nanofibers with a diameter of 50–80 nm and a length of 400 nm–600 nm. Figure 2c is the XRD diagram before and after the modification of TiO_2_. The diffraction peaks of the samples were matched with the standard card of TiO_2_ (JCPDS Card No. 21-1272), which indicates that the TiO_2_ nanofibers were mainly the anatase type. The diffraction peaks at 25.5°, 37.7°, 47.9°, 53.7°, 55.1°, and 62.8°, respectively, corresponded to the (101), (004), (200), (105), (211), and (204) crystal planes of anatase TiO_2_ crystals [27,28].

### 3.2. PDMS/TiO_2_-Coated Stainless Steel Mesh

SEM analysis was carried out to examine the surface morphology. The change in morphology after coating is shown in Figure 3. As seen in Figure 3a, the uncoated stainless steel mesh had a relatively smooth surface. After coating with the PDMS/TiO_2_ composite coating (Figure 3c,d), the stainless steel wires were covered uniformly and completely by the coating, and a relatively rough surface was observed, while the structure of the stainless steel mesh was well preserved. The diameter of the uncoated stainless steel wire was calculated to be 26.9 ± 2.4 μm, while the diameter of the coated stainless steel wire was 31.3 ± 4.7 μm. The thickness of the composite coating was then calculated to be about 2.2 ± 1.2 μm. The composite coating covered completely the stainless steel wires, and almost no coating was observed in the pores of the coated stainless steel mesh, which guaranteed the free flow of the water–oil mixture.

### 3.3. Contact Angle Measurement

The surface wettability of the coated SSM was characterized by water contact angle, as shown in Figure 4. The contact angle of the pretreated SSM was almost 0 (Figure 4a), and the water droplets rapidly penetrated. The composite coating gave the SSM superhydrophobic propertidx, and the contact angle reached 155° ± 0.5° (Figure 4b). Figure 4c,d show the influence of the pH on contact angle. The contact angle decreases slightly in the range of pH value from 1 to 14, but is greater than 150°. The water droplets with pH of 1, 7, and 14 on the superhydrophobic coating are perfectly spherical. When different liquids were dropped onto the coated SSM, the red n-hexane droplets rapidly permeated, leaving only the red oily imprint. Other liquids showed the morphology of spherical droplets on the surface (see the photograph of Figure 4e), demonstrating that the materials treated with composite coating have good hydrophobic property to different aqueous solutions.

### 3.4. Corrosion Resistance Test

It is necessary to consider the anti-corrosion property of stainless steel mesh during use. Electrochemical impedance spectroscopy (EIS) was used to study the corrosion resistance of the SSM. Figure 5a shows the Nyquist plots of the pretreated stainless steel electrode and the coated stainless steel electrode after soaking in 3.5% NaCl solution for 1 h. In the initial immersion, the aggressive ions unevenly penetrated into the coating and did not reach the coating/steel interface; both electrodes had a capacitance arc, and the electrode after coating had a larger capacitance arc than the blank electrode, indicating that the coated stainless steel electrode had higher corrosion resistance. As time progressed, the corrosive ions in the coating became homogeneous. There was an electrocoupling layer capacitor on the corrosive interface, which formed a parallel circuit with the interface resistance. At this stage, the coating had not been completely destroyed and it could still exhibit anticorrosion performance. After 30 days of immersion (Figure 5b), it was found that the capacitive resistance was reduced, but it was still far greater than that of the blank stainless steel electrode, indicating that the prepared composite coating has a long-term protective effect on stainless steel. With the prolongation of the immersion time, the accumulation of corrosion products at the interface between the coating and mild steel resulted in the appearance of Warburg diffusion resistance. Warburg impedance appeared in the low-frequency region, indicating that the diffusion of ions appeared in the coating, and the protective effect of the coating was weakened. The results showed that the surface of the composite coating had good anti-corrosion performance and could be used in corrosive environments.

### 3.5. Oil–Water Separation Performance

In order to study the oil–water separation performance of the coated SSM, several oils such as chloroform, n-hexane, vegetable oil, and mineral oil were used for separation tests. As shown in Figure 6, the separation efficiency of the coated SSM for different oils was above 94%. The separation efficiency of n-hexane (94.8%) and chloroform (94.1%) was slightly lower than that of other oils due to the high volatility; in addition, viscosity was another factor affecting the separation efficiency. Therefore, vegetable oil (96.3%) remained on the inner wall and the mesh of the separation device, resulting in a low separation efficiency [29]. When the water phase was changed to aqueous solutions with different pH values, the separation efficiency basically did not change; thus, it was shown that the coated SSM could effectively separate different oil–water mixtures. Table 1 shows a comparison of the separation between the TiO_2_/PDMS-coated SSM and other separation materials reported in the references.

### 3.6. Durability Measurement

One of the most important aspects of oil–water separating meshes is their durability against harsh mechanical conditions. In order to investigate the durability of the coated SSM, a wear resistance test was carried out. As shown in Figure 7a, after 40 cycles of sanding, the surface still maintained its superhydrophobicity, which verified its excellent wear resistance and durability.

The reusability performance of the coated SSM was studied according to the separation efficiency of the n-hexane/water mixture, as shown in Figure 7b. It was clear that even after 30 times of separation, the superhydrophobic SSM still showed excellent separation efficiency of more than 94% and a contact angle of 150.4°. However, with the increase of the number of separation cycles, the separation efficiency decreased, but it was still above 92%. By observing the separation device, it was found that the main reason for the decrease in efficiency was that a small amount of n-hexane volatilized, which led to the reduction of the collected n-hexane and of the separation efficiency. The results showed that the superhydrophobic SSM had stable oil–water separation efficiency and good reusability.

### 3.7. Self-Cleaning Performance

The self-cleaning performance of the composite coating was measured by cleaning soil. A small amount of soil was sprinkled on the coated glass slide, which was placed in a petri dish to form a small inclination, as shown in Figure 8a. The water dripped onto the tested glass slide through the syringe. During the water drop falling test, the soil was wrapped and quickly removed by the falling water drops so as to obtain a clean surface, as shown in Figure 8b,c. This is because the prepared superhydrophobic coating had a very small sliding angle. Under the impact pressure and injection direction of the water droplets, it was easy for the water droplets roll down and carry away the pollutants on the surface. Even if the pollutants were wrapped, the spherical form of the liquid water droplets could be maintained. Here, the sliding angle of the coated glass is small, and accordingly the soil on the coated glass is easy to be removed as the water drops roll down, as shown in Figure 8d.

The reason why keeping the superhydrophobic coatings stable is important is that the powerful water jet could be damaging to the coating of a surface as much as the former could make the latter’s wettability decline [37,38]. According to this fact, this study revealed how to use a water jet impact as a way to test its ability to keep hybrid microsphere coatings stable. With an angle of incidence and a high speed, a blue colored water jet, about 3 cm above the sample surface, was syringed toward the superhydrophobic coating surface. Reflected in a direct way at an angle of about 20° from the contact point, Figure 9a revealed that the water jet didnot spread on the surface, neither did the superhydrophobic coating change. The coating could keep the appearance original as much as 10 min after the water jet test. What’s more, as it could be observed from the image of SEM in Figure 9b, it was the hierarchical micro/nano structure of the coating that remained almost intact. The CA keep 150° unchanged even after the test. Thus, we could conclude that the superhydrophobic coating composed of the TiO_2_ fiber held high stability toward the water jet.

## 4. Conclusions

(1)The TiO_2_ nanofibers prepared by the hydrothermal method are uniform in size. The TiO_2_ nanofibers are added to the PDMS to form a superhydrophobic coating, which is sprayed on the stainless steel mesh to obtain good superhydrophobic performance. At the same time, the superhydrophobic coating has excellent corrosion resistance and can still protect the stainless steel mesh after 720 h of immersion;(2)When the superhydrophobic SSM is used as oil–water separation material, the separation efficiency of different oil–water mixtures is above 94%, and the separation efficiency is basically unchanged in acid, alkali, and salt environments;(3)After 50 times of continuous separation, the separation efficiency is still above 92%. After 40 cycles of polishing, the coating still has a high water contact angle, indicating that the superhydrophobic coating has good wear durability;(4)The PDMS/TiO_2_ composite coating has excellent self-cleaning performance.

## Figures and Tables

**Figure 1 polymers-14-05431-f001:**
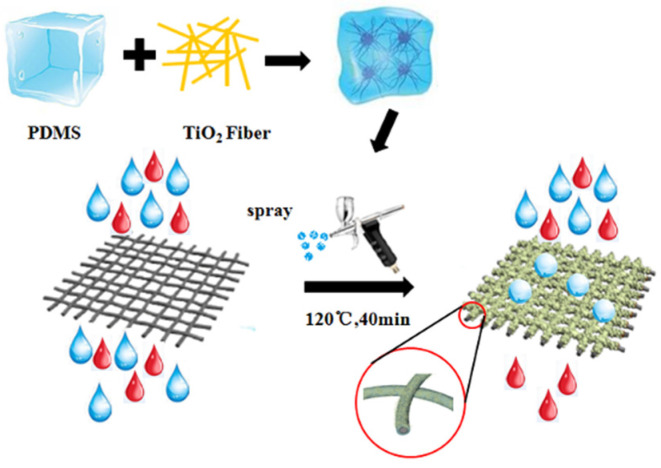
Sketch of preparation of the superhydrophobic stainless steel mesh.

**Figure 2 polymers-14-05431-f002:**
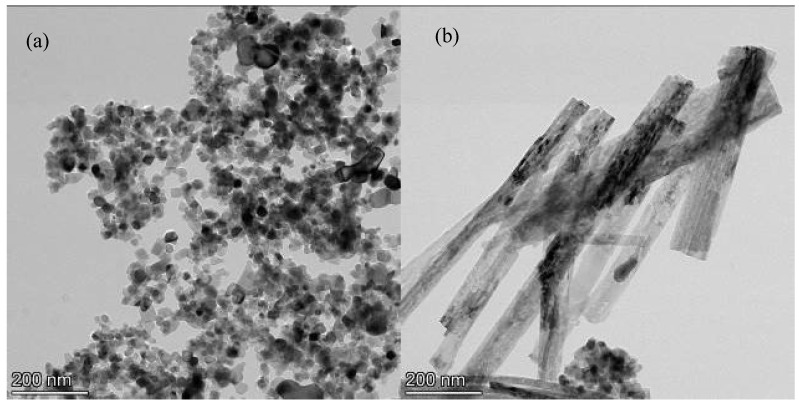
Morphological and compositional analysis of TiO_2_. TEM of P25 (**a**) and TiO_2_ fiber (**b**), and XRD of P25 and TiO_2_ fiber (**c**).

**Figure 3 polymers-14-05431-f003:**
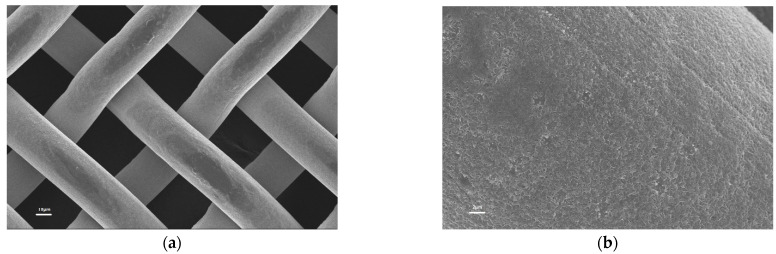
SEM images of (**a**,**b**) uncoated and (**c**,**d**) coated mesh.

**Figure 4 polymers-14-05431-f004:**
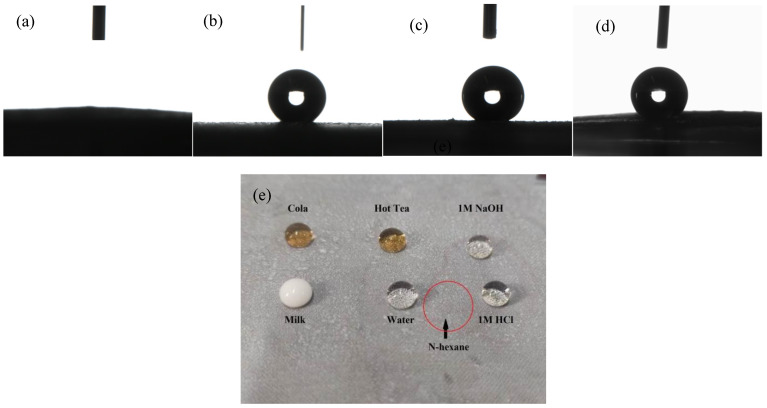
Variation of water contact angles on the superhydrophobic coating composed of PDMS/TiO_2_ fiber at pH values ranging from 1 to 14: pretreated mesh (**a**), pH = 7 (**b**), pH = 1 (**c**), pH = 14 (**d**), and the photograph of different liquids on the SSM (**e**).

**Figure 5 polymers-14-05431-f005:**
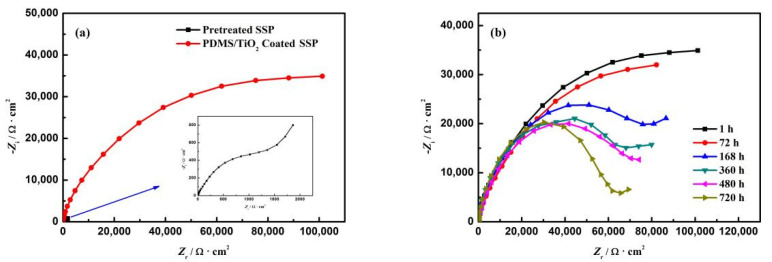
Nyquist plots of the coated mesh: after 1 h immersion (**a**) and a long-term test (**b**).

**Figure 6 polymers-14-05431-f006:**
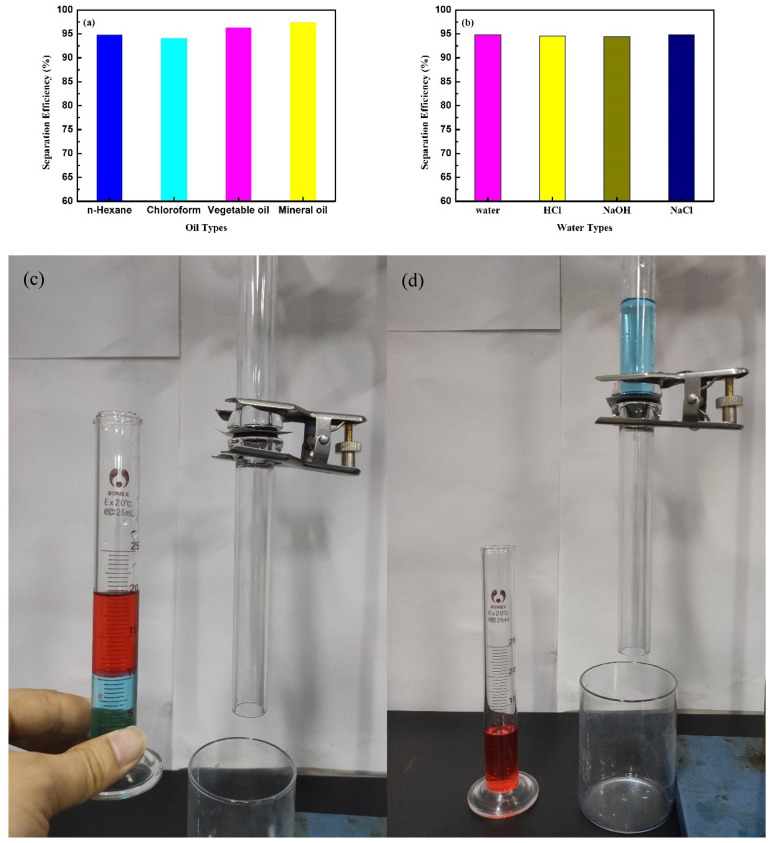
Oil–water separation efficiency and device of coated mesh: different oil–water mixture (**a**), oil–water mixture with different pH values (**b**), the device of oil–water separation (**c**,**d**).

**Figure 7 polymers-14-05431-f007:**
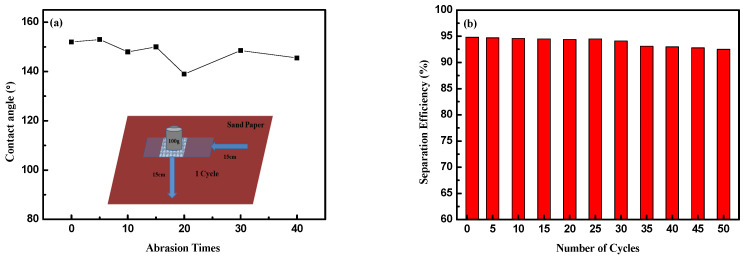
Durability of the coated mesh: wear resistance (**a**) and recycling separation (**b**).

**Figure 8 polymers-14-05431-f008:**
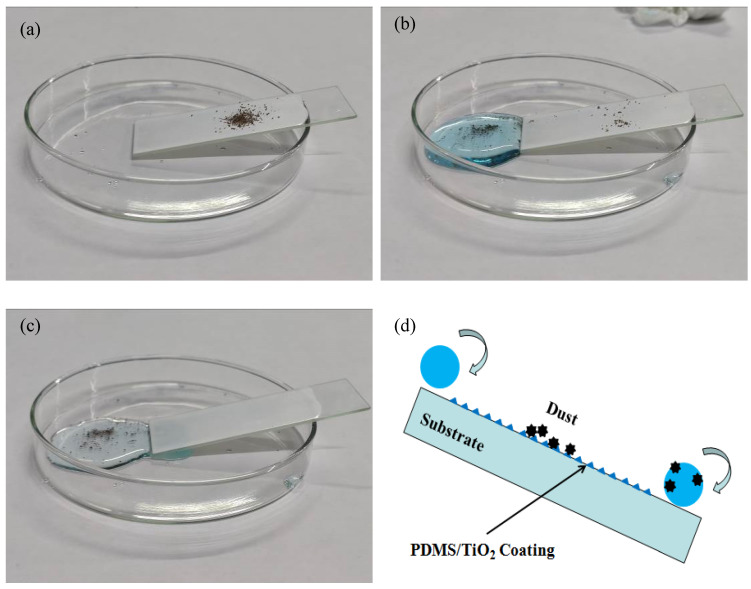
Self-cleaning properties of the PDMS/TiO_2_ coating: (**a**) before (**b**) during and (**c**) after water droplet cleaning; (**d**) Schematic illustration of the self-cleaning process.

**Figure 9 polymers-14-05431-f009:**
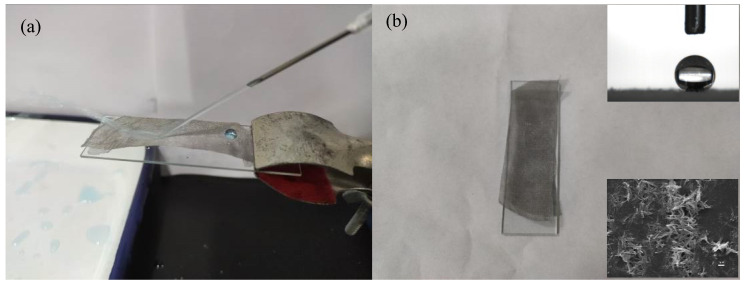
Test of water jet stability for the coated mesh: (**a**) During water jet progress; (**b**) The appearance after 10 min water jet test. The inserted SEM image shows the microstructure of the super-hydrophobic coating after the test. The upper right inset is the optical photograph for the water contact angle measurement after the water jet test.

**Table 1 polymers-14-05431-t001:** Comparison of the SSM and some reported separation materials.

Materials	Base Materials	Oil	Separation Efficiency (%)	References
TiO_2_ nanoparticles + PDMS	Iron–chromium–nickel alloy meshes	Hexane	95	[24]
SiO_2_ + PVDF	Filter paper	Dichloromethane	97.5	[30]
Magnesium stearate + phenol formaldehyde resin	SSM	Hexane	95	[31]
TiO_2_/SiO_2_ + epoxy resin	SSM	Soybean oil	95	[32]
TiO_2_ + propyl methacrylate	Cotton fabrics	Dichloromethane	96	[33]
Nanoscale SiO_2_ + dodecyl trimethoxy silane	SSM	Kerosene	95	[34]
Docosanethiol + octavinylsilsesquioxane	PET fabrics	Toluene	97	[35]
F-HNTs + epoxy resin	SSM	Dichloromethane	97.8	[36]
TiO_2_ nanofibers + PDMS	SSM	Hexane	95	This work

## Data Availability

The data supporting this study are available from the corresponding author upon reasonable request.

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
