# Peer review of "Application of Superhydrophobic Mesh Coated by PDMS/TiO2 Nanocomposites for Oil/Water Separation"

_polymers, 2022, doi:10.3390/polym14245431_

Round 1
Reviewer 1 Report
The authors present a study of the application of PDMS-TiO2 nanocomposites onto SS mesh and its incidence in some properties such as anticorrosion resistance, and oil-water separation performance.
The resulting samples are superhydrophobic and self-cleaning. The samples are well characterized by TEM, SEM, and XRD.
As compared with the literature, the contribution of the intended manuscript is modest. For example, there are several studies of the PDMS-TiO2 nanocomposites onto SS mesh:
Wang et al. Superhydrophobic and photocatalytic PDMS/TiO2 coatings with environmental stability and multifunctionality. doi 10.1016/j.colsurfa.2018.10.054
Zhang et al. Self-cleaning superhydrophobic surface based on titanium dioxide nanowires combined with polydimethylsiloxane. doi: 10.1016/j.apsusc.2013.07.100
Qing et al. Facile fabrication of superhydrophobic surfaces with corrosion resistance by nanocomposite coating of TiO2 and polydimethylsiloxane. doi: 10.1016/j.colsurfa.2015.08.024
Zhou et al, Fabrication of superhydrophobic PDMS/TiO2 composite coatings with corrosion resistance. 10.1680/jsuin.22.00013
Bolvardi et al. Towards an efficient and durable superhydrophobic mesh coated by PDMS/ TiO2 nanocomposites for oil/water separation. doi 10.1016/j.apsusc.2019.06.268
From this point of view, what is the relevance of the study presented by the authors?
Some minor mistakes like typographical errors and errors in the values communicated of the size of the coated SSM must be resolved.
Could you please explain the contribution of your research to this topic?
Reviewer 2 Report
1- The title of the paper reads “Application of Superhydrophobic Mesh Coated by PDMS/TiO2
Nanocomposites for Qil/Water Separation” . I am not sure what is this “Qil/Water” separation? Wiki says “Qil is a village in Hengam Rural District, Shahab District, Qeshm County, Hormozgan Province” (see here: https://en.wikipedia.org/wiki/Qil). I am not sure what does it do with the title of the paper?
2- The introduction reads odd. It is poorly written.
3- The method section needs more elaboration since is too short. I am not sure what these authors wanted to say.
4- Results look fine, but the basic science is not properly discussed.
5- There are two pictures in Figure 4, which are not described. I see (a)-(d), but cannot see (e).
In essence, this paper requires significant language editing and further review. It cannot be acceptable in its current form.
Round 2
Reviewer 1 Report
tha uthor take on consideration all of my comments. Please acept in the actual version.
Reviewer 2 Report
The authors of this study modified their ms to some extent. But I still have a few comments on this ms. Hence I can not recommend this paper in its current form. My comments are as follows:
1. It is not clear from the introduction part, why and how this study is different from the already published reports. What is their new idea?
2. The results are discussed, but it lacks proper discussion of basic science and the validity of the obtained results. The authors must add the comparison of their results with previously published reports.
Author Response
Dear Respected Reviewer:
A lot of thanks for the comments which encouraged us to revise our manuscript again to be in a more suitable form to warrant publication in this outstanding journal. We wish that you accept our answers to the questions. Again, all your valuable comments are made (and highlighted with red or blue color within the manuscript) as follows:
1.It is not clear from the introduction part, why and how this study is different from the already published reports. What is their new idea?
Author’s Response:
Thank you for this comment.
We have add the difference in the introduction part.
2.The results are discussed, but it lacks proper discussion of basic science and the validity of the obtained results. The authors must add the comparison of their results with previously published reports.
Author’s Response:
Thank you for your comment.
We have add the comparison with the previously published papers in section 3.6.
Round 3
Reviewer 2 Report
Authors of this work have revised their ms based on my previous review. The introduction part, together with picture qualities, seems improved. The results and discussion part is also understandable. I have no problems if this work can be acceptable by the editorial office.